# A Rolling Bearing Fault Diagnosis Method Based on the WOA-VMD and the GAT

**DOI:** 10.3390/e25060889

**Published:** 2023-06-01

**Authors:** Yaping Wang, Sheng Zhang, Ruofan Cao, Di Xu, Yuqi Fan

**Affiliations:** 1Key Laboratory of Advanced Manufacturing and Intelligent Technology, Harbin University of Science and Technology, Harbin 150080, China; wangyaping@hrbust.edu.cn; 2School of Mechanical and Power Engineering, Harbin University of Science and Technology, Harbin 150080, China; cxzzxc147258@163.com (S.Z.); kk837910389@163.com (R.C.); dixu115@hrbust.edu.cn (D.X.)

**Keywords:** fault diagnosis, signal decomposition, GAT, rolling bearing, WOA-VMD

## Abstract

In complex industrial environments, the vibration signal of the rolling bearing is covered by noise, which makes fault diagnosis inaccurate. In order to overcome the effect of noise on the signal, a rolling bearing fault diagnosis method based on the WOA-VMD (Whale Optimization Algorithm-Variational Mode Decomposition) and the GAT (Graph Attention network) is proposed to deal with end effect and mode mixing issues in signal decomposition. Firstly, the WOA is used to adaptively determine the penalty factor and decomposition layers in the VMD algorithm. Meanwhile, the optimal combination is determined and input into the VMD, which is used to decompose the original signal. Then, the Pearson correlation coefficient method is used to select IMF (Intrinsic Mode Function) components that have a high correlation with the original signal, and selected IMF components are reconstructed to remove the noise in the original signal. Finally, the KNN (K-Nearest Neighbor) method is used to construct the graph structure data. The multi-headed attention mechanism is used to construct the fault diagnosis model of the GAT rolling bearing in order to classify the signal. The results show an obvious noise reduction effect in the high-frequency part of the signal after the application of the proposed method, where a large amount of noise was removed. In the diagnosis of rolling bearing faults, the accuracy of the test set diagnosis in this study was 100%, which is higher than that of the four other compared methods, and the diagnosis accuracy rate of various faults reached 100%.

## 1. Introduction

Rolling bearings are widely used in various fields of the machinery industry due to their advantages of high speed, high efficiency and low noise. They are often used in harsh operating environments, which result in the large dispersion of life and a high failure rate. According to statistics, about 30% of mechanical failures in rotating machinery equipment that use rolling bearings are related to bearing damage [1]. The use of vibration signals generated during the working process for fault diagnosis may not only reduce the probability of accidents related to mechanical equipment, but it can also provide reliable decision support for the later maintenance of the equipment [2,3]. As fault detection is based on physical quantities (such as stator current [4], stray flux [5], thermal image [6], etc.), and because vibration-based measurement methods are lower in cost, are easier than direct observation, more sensitive to external interference, and often used in practical engineering, a vibration-based fault detection method is adopted in this paper.

In general, the vibration signal collected through the equipment is mixed with a considerable amount of noise. Due to the existence of noise, the performances of the mechanical equipment and equipment health prediction are very poor. To achieve optimal noise reduction, it is particularly important to select a suitable noise reduction method. The accuracy of fault diagnosis depends to a certain extent on the noise reduction effect. For the problem of noise reduction, Chen applied EMD (Empirical Mode Decomposition) to process the original signal containing the interference signal and then further carried out a feature extraction process [7]. The extracted time-domain and frequency-domain features formed a new rolling bearing fault feature set. Zhou solved the strong noise in bearing vibration signals by combining the wavelet threshold with EMD, which achieved good results [8]. Although the above noise reduction methods are all effective, they also have certain drawbacks, such as mode mixing and end effect in the decomposition process [9,10]. Jia decomposed the collected rolling bearing vibration signals using EEMD (Ensemble Empirical Mode Decomposition) to effectively remove the noise [11]. Niu also used EEMD for noise reduction, which did not completely remove the interference signal in the vibration signal as compared to EMD and increased the calculation time [12]. Donoho first proposed the wavelet transform method, which adopted soft and hard thresholds to filter the wavelet coefficients in the decomposition process in order to achieve noise reduction [13]. Liang proposed a new method, the WT-IResNet (Wavelet Transform-Improved Residual Neural Network), for signal noise reduction [14]. Among the current signal noise reduction methods, it is wavelet transform that has a good effect [15]. Threshold setting is a key step in the wavelet decomposition process [16], which is usually selected by means of experimentation or manual experience. Since it cannot be adjusted according to different signals, the model is poorly adaptive. The VMD solves to a certain extent the problems of mode mixing and end effects, and it can effectively separate components and achieve the adaptive frequency-domain separation of the signal, mainly by obtaining the bandwidth and frequency center of each IMF [17]. Duo used the VMD method for the problem of bearings with high external environmental interference and a large noise component [18]. Wu proposed an independent component (ICA) algorithm based on the VMD because the fault signal generated in the gearbox was very weak and easily affected by external environmental noise and other factors [19]. It was proposed that two key parameters, the number of mode decompositions and the quadratic penalty factor, need to be set artificially before the VMD decomposition. However, this artificial setting is subjective and poorly generalized. Therefore, how to improve the choice of these two key parameters is very important for the subsequent diagnosis of rolling bearing faults.

After the appearance of new technology for rolling bearing condition detection and fault diagnosis, new ideas have constantly emerged with the progress of the times [20]. Rolling bearing fault diagnosis technology has also entered the peak of development [21]. Gao and Li used a convolutional neural network for the fault diagnosis of rolling bearings, and their results showed that it had high recognition accuracy for different types of faults [22,23]. Ma proposed a fault diagnosis method based on complementary integrated empirical mode decomposition combined with principal component analysis and limit gradient lifting; the method has been verified to be effective [24]. However, these fault diagnosis methods also have some limitations, and do not explore the relationships and interdependences of data. To solve this problem, some studies proposed to display data in the form of irregular plots [25,26,27,28]. Compared with traditional methods, it is necessary to establish graph data first, which increases the degree of complexity and poses severe challenges to standard neural network-based methods. It makes some important operations (e.g., convolution) easy to apply to the Euclidean domain, but it is difficult to model the graph data in non-Euclidean space [29]. A graph neural network (GNN) [30] is an artificial intelligence algorithm derived from graph theory, that can process graph data. Due to the ability of the GNN to model the interdependence between data and embed it into the extracted features, this method has gradually become a research hotspot in the field of rolling bearing fault diagnosis. Gao utilized all vibration samples to construct a undirected weighted K-nearest neighbor graph and also used a depth graph neural network for fault diagnosis. The effectiveness of this method was then verified with gear and bearing data [31]. Li proposed a graph convolutional network (GCN) combined with a weighted horizontal visibility graph (WHVG) for bearing fault diagnosis [32]. The WHVG is used to convert a time series into graphical data from a geometric perspective, which improves the graph isomorphic network (GIN) to GIN+, while learning graph representation and fault classification. Finally, the effectiveness of the WHVG and GIN+ was verified with three actual bearing datasets. In another study, Zhang transformed acoustic signals into a graph and modeled the graph using a GCN to carry out the fault diagnosis of roller bearings [33]. Yu first constructed a graph dataset and then realized fault classification using a fast deep GCN [34]. Li transformed the vibration signals of rolling bearings into horizontal visibility graphs, and then modeled the graph data with a GNN to realize fault classification [35]. The above methods ignore the importance of input information and the interdependence of data. For different neighborhoods, the fault information is related to different degrees. Therefore, if the same weight is given, a certain degree of information will be lost, which will affect the outcome of the rolling bearing fault diagnosis.

Hence, a rolling bearing fault diagnosis method based on the WOA-VMD and GAT is proposed in this paper. The main contributions of this method are as follows:By separating the signals with a fixed bandwidth, the problems of mode mixing and end effect are solved to some extent. Two key parameters in the VMD are determined using the WOA optimization algorithm, allowing the model to adaptively decompose the signal.Node classification of graph structure data is carried out using the attention mechanism method, which assigns different attention weights to different neighborhoods so as to identify more important information.

## 2. Signal Decomposition and Reconstruction Based on the WOA-VMD

The variational mode decomposition can be estimated for each signal component by solving a frequency-domain variational optimization problem. It is assumed that the decomposed IMF components are all low-bandwidth signals, and that the fault feature frequency appears in the central frequency of the IMF components. The IMF components are then reconstructed to achieve the purpose, the VMD model is solved iteratively. The center frequency $\omega_{k}$ is used to decompose the decomposed k-mode components. Finally, the following problems are obtained:(1)$$\begin{array}{c} \underset{\left\{u_{k}\right\},\left\{\omega_{k}\right\}}{min}\left\{\sum_{k}\|\partial_{t}\left[\left(\delta \left(t\right)+\frac{j}{\pi t}\right)\times u_{k}\left(t\right)\right]e^{-j\omega_{k}t}{\|}_{2}^{2}\right\}\\ s.t\sum_{k}u_{k}=f \end{array},$$
where $\left\{u_{k}\right\}=\left\{u_{1},\cdots {,u}_{k}\right\}$, $u_{k}$ is the original signal, $u_{1},\ldots ,u_{k}$ are the k mode components obtained after decomposition, and $\left\{\omega_{k}\right\}=\left\{\omega_{1},\cdots ,\omega_{k}\right\}$ is the frequency center of each mode component after decomposition. If Equation (1) is to be solved, it needs to be converted into the unconstrained problem of Equation (2). The conversion mode is mainly solved with the use of Lagrange multipliers and quadratic penalty terms.
(2)$$L\left(\left\{u_{k}\right\},\left\{\omega_{k}\right\},\lambda \right)=\alpha \sum_{k}\|\partial_{t}[(\delta \left(t\right)+\frac{j}{\pi t})\times u_{k}\left(t\right)]e^{-j\omega_{k}t}\|+\|f\left(t\right)-\;\sum_{k}u_{k}\left(t\right){\|}_{2}^{2}+\langle \lambda \left(t\right),f\left(t\right)-\sum_{k}u_{k}\left(t\right)\rangle ,$$
where α is the penalty factor, and $\lambda \left(t\right)$ is the Lagrange multiplier.
(3)$$\omega_{k}^{n+1}=\frac{\int_{0}^{\infty }\omega {\left|u_{k}\left(\omega \right)\right|}^{2}d\omega }{\int_{0}^{\infty }{\left|u_{k}\left(\omega \right)\right|}^{2}d\omega },$$

The two parameters of the VMD, the penalty factor α and the decomposition layers k, need to be considered in the decomposition process to create certain limitations for the VMD. Too small or too large a value will affect the algorithm’s effect, so the optimal parameter combination [k, α] needs to be determined. At present, the center frequency observation method is widely used. This method mainly determines the value of k by observing the center frequency under different values of k, without an accurate basis, and it can only determine the number of modes k, but not the penalty parameter α, which ultimately leads to poor noise reduction.

### 2.1. Whale Optimization Algorithm

In this paper, the whale optimization algorithm is used to adaptively determine the two parameters mentioned above in order to achieve a better noise reduction effect. In the process of hunting, humpback whales surround their prey in groups and move in a spiral, during which they constantly spit out bubbles, thus forming a spiral “bubble net” shown in Figure 1. This will make the space for the prey’s movement smaller and smaller, until it can be swallowed in a single bite [36].

However, the algorithm initially does not know the optimal location, and the WOA is goal-oriented to find the prey. When whales find the best prey, they also find the best whale location, thus allowing the rest of the whales to move closer to it. This behavior can be expressed as follows:(4)$$\vec{D}=\left|\vec{C}\cdot \vec{X^{*}}\left(t\right)-\vec{X}\left(t\right)\right|,$$
(5)$$\vec{X}\left(t+1\right)=\vec{X^{*}}\left(t\right)-\vec{A}\cdot \vec{D},$$
where $\vec{X}$ is the position vector, $\vec{X^{*}}$ represents the optimal position obtained currently, $\vec{A}$ and $\vec{C}$ are the coefficient vectors, t is the current number of iterations and $\vec{D}$ is the distance between the prey and the whales. If there is a better solution, $\vec{X^{*}}$ will be iteratively updated.

Vectors $\vec{A}$ and $\vec{C}$ can be calculated as follows:(6)$$\vec{A}=2\vec{a}\cdot \vec{r}-\vec{a},$$
(7)$$\vec{C}=2\cdot \vec{r},$$
where $\vec{a}$ is linear in its descent from 2 to 0, while $\vec{r}$ is a random vector within [0, 1].

Simulating the humpback whale’s unique hunting method is achieved through a spiral motion that updates the position and a narrowing ring mechanism, which is often referred to as the bubble net strategy. Assuming that the probability of prey capture using these two methods is 50%, the total strategy is expressed as follows:(8)$$X\left(t+1\right)=\left\{\begin{array}{c} \vec{X^{*}}\left(t\right)-\vec{A}\cdot \vec{D},\\ \vec{D^{\prime }}\cdot e^{bl}\cos\left(2\pi l\right)+\vec{X^{*}}\left(t\right) \end{array}\{\begin{array}{c} 0\leq p<0.5\\ 0.5\leq p\leq 1 \end{array}\right.,$$
where $\vec{D^{\prime }}=\left|\vec{X^{*}}\left(t\right)-\vec{X}\left(t\right)\right|$, b is the coefficient of the spiral search, and l is a random number within [−1, 1].

In addition to the above strategy, it will also hunt according to its position, again through a change in vector $\vec{A}$. When $\left|\vec{A}\right|>1$, it moves away from the prey; conversely, it moves closer to the prey. In this way, global optimization properties can be effectively improved. The mathematical model is as follows:(9)$$\vec{D}=\left|\vec{C}\cdot {\vec{X}}_{random}\left(t\right)-\vec{X}\left(t\right)\right|,$$
(10)$$\vec{X}\left(t+1\right)={\vec{X}}_{random}\left(t\right)-\vec{A}\cdot \vec{D},$$
where ${\vec{X}}_{random}$ is the whale position randomly selected from the current group of whales.

Then, the minimum value of envelope entropy is selected as the fitness function; the envelope entropy represents the sparsity of the original signal. When there is more noise in the IMF and less effective information, the envelope entropy value is larger; otherwise, the envelope entropy value is smaller.

The envelope spectrum of the signal $X\left(i\right)\left(i=1,2,\ldots ,N\right)$ is calculated with the following equation:(11)$$\left.\begin{array}{c} E_{p}=-\sum_{j=1}^{N}p_{j}lgp_{j}\\ p_{j}=a\left(j\right)/\sum_{j=1}^{N}a\left(j\right) \end{array}\right\},$$
where $a\left(j\right)$ is the envelope signal of k mode components decomposed by the VMD after Hilbert demodulation, $p_{j}$ is the sequence of probability distributions obtained by calculating the normalization of $a\left(j\right)$, N is the number of sampling points, and the entropy value of a calculated sequence of probability distributions $p_{j}$ is the envelope entropy $E_{p}$.

### 2.2. WOA-VMD Parameter Optimization

Since the WOA is simple, and easy to implement and because few parameters were set, it was decided that the penalty factor and decomposition layers in the VMD decomposition would be optimized first. The steps are shown in Figure 2.

(1)Set the number of whales, the maximum number of iterations and the optimization dimension, and initialize the position information. Set the mode component and penalty factor as $k=\left[100,\;3\right]$ and $\alpha =\left[2000,\;7\right]$;(2)Use the VMD algorithm to decompose the input signal and obtain each IMF function. Calculate the envelope entropy of each IMF according to Equation (11). The envelope entropy was used as a fitness function to find the optimal whale location and retain it;(3)Start the iteration. Generate a random number *p* in the interval (−1, 1). If p<0.5, it is directly transferred to step 4; otherwise, Equation (8) is used for position update, namely for spiral contraction;(4)Determine the value of $\left|A\right|$. If $\left|A\right|<1$, update the position type of Equation (5), surrounded by the contraction; otherwise, update the position according to Equation (10), that is, change it to random exploration;(5)Calculate the fitness of each whale and compare it with the previously reserved optimal position. If it is better, replace it with the new optimal solution;(6)Determine whether the iteration is terminated. If $t\leq t_{max}$, then t=t+1; return to Step 3. Otherwise, the iteration ends and the optimal parameter combination $\left[k,\alpha \right]$ is saved.

### 2.3. Screening of IMF Component Coefficients

To filter the decomposed mode components, the Pearson correlation coefficient method is adopted in this paper.

#### 2.3.1. Pearson Correlation Coefficient Analysis

The Pearson correlation coefficient method focuses on the correlation between signals to determine the similarity between two signals. The Pearson correlation coefficient method is as follows:(12)$$\rho_{x_{t}x_{IMF}}=\frac{\mathrm{cov}(x_{t},x_{IMF})}{\sigma_{x_{t}}\sigma_{x_{IMF}}}=\frac{E\left(x_{t}-E\left(x_{t}\right)\right)\left(x_{IMF}-E\left(x_{IMF}\right)\right)}{\sigma_{x_{t}}\sigma_{x_{IMF}}},$$
(13)$$\sigma_{x_{t}}=\sqrt{E\left(x_{t}{}^{2}\right)-E^{2}\left(x_{t}\right)},$$
(14)$$\sigma_{x_{IMF}}=\sqrt{E\left(x_{IMF}{}^{2}\right)-E^{2}\left(x_{IMF}\right)},$$
(15)$$cov\left(x_{t},x_{IMF}\right)=E\left(x_{t}x_{IMF}\right)-E\left(x_{t}\right)E\left(x_{IMF}\right),$$
where $\rho_{x_{t}x_{IMF}}$ represents the overall correlation coefficient, *E* represents the expected value, and $cov(x_{t},x_{IMF})$ represents covariances of $x_{t}$ and $x_{IMF}$.

The equation for the correlation coefficient value $r\left(x_{t},x_{IMF}\right)$ is as follows:(16)$$r\left(x_{t},x_{IMF}\right)=\frac{\sum (x_{t}-\bar{x_{t}})\left(x_{IMF}-\bar{x_{IMF}}\right)}{\sqrt{\sum {\left(x_{t}-\bar{x_{t}}\right)}^{2}}\sqrt{\sum {\left(x_{IMF}-\bar{x_{IMF}}\right)}^{2}}},$$

#### 2.3.2. Correlation Component Discrimination

The value range of $r\left(x_{t},x_{IMF}\right)$ is [−1, 1]; that is, $-1\leq r\left(x_{t},x_{IMF}\right)\leq 1$. Therefore, the classification of $r\left(x_{t},x_{IMF}\right)$ is shown in Table 1.

In this section, reference [37] is used to select the mode components. In general, it is considered that the mode components with a strong correlation or above the original signal are those with less noise information. Mode components with correlation degrees below a strong correlation are removed, and then, mode components with a strong correlation or above are reconstructed to complete the noise reduction.

The signal decomposition and reconstruction method proposed in this paper uses the VMD to decompose signals, which can effectively avoid the phenomenon of mode mixing in noise reduction with the traditional method, and can retain the effective information of the original signal. Then, two key parameters, k and α, of the VMD are determined adaptively using the WOA optimization algorithm to improve the generalization ability of the model.

This method is used for noise reduction. The parameters are initialized for the WOA. The envelope entropy value corresponding to each whale is calculated and recorded as optimal. The optimal combination of parameters is output, and the optimal k and α are used to decompose the original signal. Pearson is then used to filter the obtained results. Specific steps are shown in Figure 3.

## 3. Fault Diagnosis of Rolling Bearing Based on the GAT

If the fault can be found in time during the actual operation of rolling bearings and maintenance or remedial measures can be taken, which is of great significance for the reliability of the bearing operation. In this section, fault diagnosis methods of rolling bearings are studied based on Section 2. The GAT network model is established to diagnose rolling bearings. Meanwhile, a multi-head attention mechanism is used to perform node classification of graph structure data. To investigate ways of combining the two methods, increasing the weight of important information and improving diagnostic accuracy. At the same time, the parameter settings and data sets in the experiment are elaborated, and the experimental results are analyzed in detail.

### 3.1. KNN Graph Construction Method for Fault Diagnosis Data

Graph attention neural networks need to build graphs to represent the correlation between different faults as input data. Therefore, for the data set V after noise reduction using the WOA-VMD, the concept of Graph is introduced in order to represent the data itself and the relationship between them. Additionally, the KNN method is used to construct the graph model.

In the KNN figure, for each node (e.g., fault data point in V) to find the first K nearest neighbor points, the nearest neighbor of the obtained node $x_{i}$ can be expressed as follows:(17)$$Ne\left(x_{i}\right)=KNN\left(K^{\prime },x_{i},\psi \right),$$
where *NN* returns the K nearest neighbors of the node $x_{i}$ in set Ψ, $\Psi =\left[x_{i+1},x_{i+2},\ldots ,x_{i+m}\right]$ denotes that there are m samples, and $Ne\left(x_{i}\right)$ represents the neighbors of the node $x_{i}$.

The edge weight between *KNN* nodes can be estimated using the Gaussian kernel weight function, and defined as follows:(18)$$e_{ij}=\exp\left(-\frac{{\left(\left\|x_{i},x_{j}\right\|\right)}^{2}}{2\xi }\right),x_{j}\in Ne\left(x_{i}\right),$$
where $e_{ij}$ is edge weight between nodes $x_{i}$ and $x_{j}$, and ξ is the bandwidth of the Gaussian kernel. An example KNN composition is shown in Figure 4 [29].

The rolling bearing fault diagnosis data in this paper are based on PyTorch Geometry (PYG) [38], a deep learning framework developed by PyTorch to construct undirected graphs. Firstly, nine different types of data from Case Western Reserve University are input, the number of labels is set, the time-domain vibration signal is loaded as the input, the graph model is constructed using *KNN*, and weights are assigned. Then, PYG is used for data encapsulation to complete the establishment of the graph model. The composition process is shown in Figure 5.

### 3.2. Graph Attention Layer Construction Method

This section will start from the attention layer of a single graph. Firstly, node information in the constructed graph model is input, and then input features are transformed into higher-order features via a linear transformation. In addition, attention (weights) is allocated to each node through self-attention; a represents the shared attention mechanism, and $R^{F^{\prime }}\times R^{F^{\prime }}\to R$, which is used to calculate the attention coefficient $e_{ij}$, that is, the influence coefficient of node i on node j.
(19)$$e_{ij}=a\left(W{\vec{h}}_{i},W{\vec{h}}_{j}\right),$$

The attention calculation above only considers any two nodes in the graph, whereas in the general case, every node in the graph needs to be considered; thus, the overall graph information is lost. Therefore, this section only calculates the correlation between node i and node $j\in N_{i}$ in its neighborhood to the target node, where $N_{i}$ is a domain of node i, and then normalizes it in all j options with the softmax function.
(20)$$\alpha_{ij}=softmax_{j}\left(e_{ij}\right)=\frac{exp(e_{ij})}{\sum_{k\in N_{i}}exp(e_{ik})},$$

Using *LeakyReLU* as the activation function.
(21)$$e_{ij}=LeakyReLU({\vec{a}}^{T}[W{\vec{h}}_{i}∥W{\vec{h}}_{j}]),$$
where ∥ indicates the splicing operation, and T indicates transposition. The complete equation for calculating the weight factor is shown below.
(22)$$\alpha_{ij}=\frac{LeakyReLU({\vec{a}}^{T}[W{\vec{h}}_{i}∥W{\vec{h}}_{j}])}{\sum_{k\in N_{i}}exp(LeakyReLU({\vec{a}}^{T}[W{\vec{h}}_{i}∥W{\vec{h}}_{k}]))},$$

The attention coefficient after normalization is calculated for the corresponding linear combination of features, and the final output feature vector of each node is obtained after calculation via the nonlinear activation function as shown below.
(23)$$\vec{h_{1}^{\prime }}=\sigma \left(\sum_{j\in N_{i}}\alpha_{ij}W{\vec{h}}_{j}\right),$$

In addition, in this section, weights are mainly assigned through a multi-head attention mechanism, which is detailed in Figure 6. Each head attention mechanism will finally do a summation and averaging process to $\vec{h_{1}^{\prime }}$ [39].

The output of the above equation is stitched together using the following independent attention mechanisms:(24)$$\vec{h_{i}^{\prime }}={∥}_{\beta =1}^{K}\sigma (\sum_{j\in N_{i}}\alpha_{ij}^{k}W^{k}{\vec{h}}_{j}),$$
where ∥ represents the splicing operation, $\alpha_{ij}^{k}$ represents the normalized attention coefficient calculated by the β-th attention mechanism, and $W^{k}$ is the weight matrix of the corresponding input linear transformation. It is worth noting here that, in this setup, the final output $h^{\prime }$ returned will consist of the K and $F^{\prime }$ features of each node, not just $F^{\prime }$. The specific flow is shown in Figure 7.

### 3.3. Formatting of Mathematical Components

In this section, the graph attention neural network model is constructed, and some notations used are introduced. The graph is represented as $G=\left\{V,E\right\}$. The graph adjacency matrix is represented as A. The node feature matrix of the graph is represented as $F\in R^{N\times d}$. N is used to represent the number of nodes in the graph, and d is used to represent the dimension of the node features. The features of a node in the graph are each a row of F.

The basic framework of the graph attention neural network is the combination of the graph filtering layer and nonlinear activation layer. Figure 8 shows two graph filtering layers and activation layers. The output of the i-th graph filter layer is denoted as $F^{\left(i\right)}$. In particular, $F^{\left(0\right)}$ is initialized to the nodal feature matrix F. The output dimension of the i-th graph filter layer is denoted as $d_{i}$. Since the structure of the graph does not change, it follows that $F^{\left(i\right)}\in R^{N\times d_{i}}$. The i-th layer of the graph filter layer can be described as follows:(25)$$F^{\left(i\right)}=h_{i}\left(A,\alpha_{i-1}(F^{i-1})\right),$$
where $\alpha_{i-1}()$ denotes the activation function applied element-by-element after the (i−1)-th graph filter layer. It is worth noting that $\alpha_{0}$ denotes a constant function; as in practice, the input features are not usually activated.

Depending on the specific downstream task, the final output $F^{\left(L\right)}$ can be used as an input for a particular layer. The downstream task is parametric learning. The GAT model in this paper takes the entire graph as inputs to generate node representations, which are then used to train a node classifier. Specifically, let $GAT_{node}()$ represent a GAT model with multiple graph filter layers stacked. The function $GAT_{node}()$ takes the graph structure and node features as inputs and the learned node features as the output, and is expressed as follows:(26)$$F^{\left(out\right)}=GAT_{node}\left(A,F,\Theta_{1}\right),$$
where $\Theta_{1}$ represents a model parameter, $A\in R^{N\times N}$ is the adjacency matrix, $F\in R^{N\times d_{in}}$ represents the node features of the input, and $F_{out}\in R^{N\times d_{out}}$ indicates the node features of the output.

The output node features are then used for node classification, as shown below:(27)$$Z=Soft\left(F^{out};\Theta_{2}\right),$$
where $Z\in R^{N\times C}$ represents the output node category probability matrix, and $\Theta_{2}\in R^{d_{out}\times C}$ is the parameter matrix that converts the feature $F_{out}$ into a dimension equal to class number C.

The i-th line of Z represents the category distribution of the prediction node, and the prediction label is usually the label with the highest probability. The whole process can be summarized as follows:(28)$$Z=f_{GAT}\left(A,F^{\left(ip\right)};\Theta \right),$$
where function $f_{GAT}()$ contains the processes of Equations (24) and (25), Θ contains $\Theta_{1}$ and $\Theta_{2}$. The parameter Θ can be learned by minimizing the following objective function:(29)$$L_{train}=\sum_{v_{i}\in V_{l}}l(f_{GAT}\left(A,F;\Theta {)}_{i},y_{i}\right),$$
where $f_{GAT}{\left(A,F;\Theta \right)}_{i}$ represents the i-th line of the matrix, that is, the category probability distribution of node $v_{i}$; $y_{i}$ represents its corresponding label; and $l\left(\cdot ,\cdot \right)$ represents some kind of loss function, such as a cross-entropy loss function.

Figure 9 shows the fault diagnosis flow chart of the GAT.

Step 1: The signal reconstruction and decomposition methods in Part A of the figure are described in Section 2.

Step 2: The GAT model is initialized and relevant model parameters are set.

Step 3: The data after noise reduction are initialized, and the graph model is constructed through the *KNN* method. The constructed graph model is divided into the test, training and validation sets.

Step 4: The training set data are input into the GAT model, the model is trained, the output error is obtained through the validation set, and the error is back-propagated to update the network model parameters.

## 4. Experimental Verification

### 4.1. WOA-VMD

In this section, the simulation fault signal and the test bench data in our laboratory are used to verify the superiority and effectiveness of the WOA-VMD noise reduction algorithm.

#### 4.1.1. Simulation Verification

Firstly, the simulation fault signal is used for verification. The equation of the simulation vibration signal is as follows:(30)$$\left\{\begin{array}{c} s_{1}=0.2\cos\left(2\pi f_{1}t+10\right)\\ s_{2}=0.4\sin\left(2\pi f_{2}t+10\right)\\ s_{3}=0.2\sin\left(2\pi f_{3}t\right) \end{array}\right.,$$
where $f_{1}=80\;\mathrm{Hz}$, $f_{2}=200\;\mathrm{Hz}$, $f_{3}=300\;\mathrm{Hz}$, sampling points N=1024.

The time and frequency domain diagrams of the simulation signal *s*_1_ are shown in Figure 10.

The time and frequency domain diagrams of the simulation signal *s*_2_ are shown in Figure 11.

The time and frequency domain diagrams of the simulation signal *s*_3_ are shown in Figure 12.

Figure 13 shows the time and frequency domain diagram of the simulation signal *Z* after mixing.

The Gaussian white noise $n\left(t\right)$ of −10 dB [40] is added to signal Y. The time and frequency domain diagrams after adding noise are shown in Figure 14.

It can be seen in Figure 14 that the simulated signal after adding noise $n\left(t\right)$ is more real than the one before. The range of variation in the time-domain diagram increases, and the difference in amplitude is likewise increased. The amplitude interleaving in the frequency domain is due to the addition of simulated high-intensity noise.

In what follows, the mixed signal with noise is input into the WOA-VMD model for decomposition, and multiple mode components and frequency-domain diagrams are obtained. The number of decomposition layers and penalty factor in the VMD algorithm is optimized using the WOA algorithm, and the best results are obtained. The whale algorithm parameters are set, as in Table 2.

The final parameters are obtained after 20 iterations. The iteration curve of penalty factor α, shown in Figure 15a, enters convergence after the fifth time, with a final convergence value of 1652. The iteration curve of the optimal decomposition layer k is shown in Figure 15b. After the second iteration, the optimal solution is found to be eight layers. The iterative curve of the envelope entropy, shown in Figure 15c, enters convergence after the thirteenth iteration, with the final envelope entropy value of 9.7633.

Figure 16 shows the time and frequency diagram of multiple mode components obtained from the decomposition of the optimal combination of parameters.

Then, the mode components obtained from decomposition are selected using the Pearson correlation coefficient method, and those whose relationship value with the original signal is greater than or equal to 0.6 are reconstructed. The correlation values of the calculated mode components are shown in Table 3.

The Pearson correlation coefficients of IMF2, 4, 5 and 6 exceed 0.6, discarding IMF1, 3, 7 and 8. IMF2, 4, 5 and 6 are reconstructed into signals after noise reduction. Then, the signals are compared after the noise reduction of EMD, EEMD, CEEMD and GA-VMD. The respective time and frequency domain diagrams are shown below.

As can be seen in Figure 17, Figure 18, Figure 19, Figure 20 and Figure 21, the WOA-VMD provides a better noise reduction effect compared to the other four methods, and it has an obvious noise reduction effect on the whole frequency band.

As can be seen in Table 4, the root mean square error after the WOA-VMD signal decomposition and reconstruction is 0.213, the signal to noise ratio is 6.912 and the noise reduction effect is more obvious as compared with other algorithms.

#### 4.1.2. Validation of Laboratory Data

As can be seen in Figure 22, the bearing fault diagnosis test bench consists of a touch panel, motor speed controller, motor, radial loading hydraulic system, ADI150 uniaxial acceleration sensor, axial loading hydraulic system, main shaft, two support 6210 and 18,720 bearing, the ER-16K bearing to be measured and a force arm beam adjusting device. The bearing type is ER-16K, and detailed parameters are shown in Table 5. The acceleration sensor was used to obtain the vibration acceleration information of 13 bearing fault states, including 10 single point faults and 3 compound faults (CF). The experimental data were obtained at a sampling frequency of 25.6 kHz. A total of 10 groups were collected under each fault state, with each group comprising 32,768 sample points.

The damage of the fault bearing is man-made, as shown in Figure 23. The inner and outer ring faults were caused by the use of a laser marking machine to create indentation in the groove of the outer ring ball. The red block represents the position of the laser groove. The rolling body fault represents the punching of holes in the rolling body.

In the experiment, as shown in Table 6, three different loads were set, and different fault locations, damage degrees and experimental speeds were set under three different loads. In addition, there were three healthy sets of bearing data for each of the three loads.

The WOA-VMD method was used for the signal decomposition and reconstruction of the original data, and the feasibility of the proposed method was observed. The time and frequency domain diagrams of the vibration signals are shown in Figure 24.

In this experiment, the inner ring fault under a 100 N load is selected as experimental data, and the sampling frequency is 1024 Hz. It can be seen in Figure 24 that there is noise in the data. The WOA-VMD is applied for noise reduction processing. The decomposed IMF components and iterative curves are shown in Figure 25 and Figure 26.

The correlation coefficient values between IMF components after the WOA-VMD decomposition and the original signal are calculated, as shown in Table 7.

After the correlation analysis, the correlation coefficients of IMF1 and 2 were higher than 0.6, so IMF1 and 2 were selected for the reconstruction, and the noise reduction process of the signal was completed. The time and frequency domain diagrams of the signal after the WOA-VMD noise reduction are shown in Figure 27.

As can be seen in Figure 27, the noise reduction effect in the high-frequency part of the signal is very obvious, with a large amount of noise having been removed. This allowed for effective information to be retained, the ideal effect to be achieved and the preparation for subsequent fault diagnosis.

### 4.2. GAT

In this section, two sets of data are chosen to verify the superiority and effectiveness of the GAT fault diagnosis algorithm. The data comprise the experimental bearing data from Case Western Reserve University [41] and the test bench data from this laboratory.

#### 4.2.1. Case Western Reserve University Data Verification

The bearing test platform of Case Western Reserve University [42] is shown in Figure 28. The fault sizes were set to 0.007 inches, 0.014 inches and 0.021 inches, with 1 inch = 2.54 cm. In addition, the rotational speeds were set to 1797 r/min, 1772 r/min, 1750 r/min and 1730 r/min. The specific different fault states are shown in Table 8.

In this section, the size of the convolutional kernel and the number of convolutional layers of the GAT will be discussed. A total of 60% of overall samples was randomly selected as the training set, 20% as the validation set and 20% as the test set. The specific GAT parameter settings are shown in Table 9.

(1)Influence of the Convolutional Kernel Size of Different Graphs

The size of the convolutional kernel Θ, an important parameter of the network model, affects the accuracy of rolling bearing fault type identification. The impact on accuracy is discussed by comparing different sizes of convolutional kernels.

As shown in Table 10, when the convolution kernel size increases from $\Theta \in R^{1024\times 1024}$ to $\Theta \in R^{2048\times 2048}$, the accuracy becomes higher, and the calculation time increases accordingly. When the convolution kernel size increases from $\Theta \in R^{2048\times 2048}$ to $\Theta \in R^{4096\times 4096}$, the calculation time increases, but overfitting occurs. Therefore, the final convolutional kernel size selected for the graph in this paper is $\Theta \in R^{2048\times 2048}$.

(2)The Effect of Different Convolution Layers

In general, the more convolutional layers there are, the more filters are superimposed to solve the learning problem hierarchically. By deepening the layers, information can be transmitted at different levels. In this section, considering that other conditions are the same, the impact of two to six convolutional layers on the diagnostic accuracy is compared, as shown in Figure 29; the loss value is shown in Figure 30.

The comparison shows that an increase in the number of layers has less impact on this experiment, but increases the time for iterations to converge. In contrast, for the loss value, the fewer the layers, the smaller the loss value, the faster the iteration speed and the more realistic the prediction. Therefore, the number of convolution kernel layers there are in this paper is two.

This time, the effectiveness of the proposed method is verified using TSNE dimensional reduction visualization, and the superiority of the GAT model is verified by comparing several fault diagnostic models.

Figure 31a shows the sample distribution of the initial data set. Before the model is trained, the label classification effect is not good. The labels of the same fault are not aggregated and the labels of different categories are mixed. It can be seen in Figure 31b that the classification of all kinds of labels was completed. The aggregation of the same category is good, and there is no mixing of different category labels. This proves that the rolling bearing fault diagnosis method proposed in this paper is effective.

As can be seen in Figure 32 and Figure 33, the accuracy of the MLP and Attention models reached about 80% after 100 training iterations, but they are not in a convergence state. Therefore, the graph neural network model converges faster and has higher diagnostic accuracy than the traditional neural network model. In the change curve of the loss value, it can be seen that the loss value of the graph neural network is lower than that of the traditional neural network.

After 100 training iterations, the accuracy of the GCN, CNN and GAT models reached about 100%, while the iteration time and loss value of the GCN model are higher than those of the CNN and GAT, and the stability of the GCN and GAT is better than that of the CNN. This indicates that the method proposed in this paper is superior, with fast iteration speed, good model stability and strong generalization ability that could better solve the problem of rolling bearing fault diagnosis.

The comparison confusion matrix obtained by the classification of the test set $V_{test}$ is shown in Figure 34. The accuracy of the experiment is calculated according to the comparison of the confusion matrix, as shown in Table 11.

The confusion matrix can analyze the classification results of samples in detail. In the figure above, the horizontal coordinates represent the predicted label of the samples, and the vertical coordinates represent the actual label of the samples. As can be seen in Figure 34, the Attention, MLP, GCN and CNN models all deviate in their diagnostic effects, while the GAT is accurate in diagnosing various faults, and its diagnostic effect is better than that of the other four methods.

It can be seen in Table 11 and Table 12 that the test set accuracy of the GAT is 100%, higher than that of the MLP, Attention, GCN and CNN models, which proves that the method proposed in this paper can realize a more accurate diagnosis of fault data.

#### 4.2.2. Validation of Laboratory Data

The experimental data of the rolling bearings obtained in the experiment were decomposed and reconstructed using the WOA-VMD signal to eliminate noise components, and were subsequently used as input. The network model and structural parameters established above were used to divide the input data. The comparative experimental results are shown in Figure 35 and Figure 36.

It can be seen in Figure 35 that the diagnostic accuracy of the MLP and Attention models are lower than that of the GCN, CNN and GAT models. It can be seen that the diagnostic accuracy of the graph structure used in this paper as input is higher, and the diagnostic accuracy of the GAT is higher than that of the GCN and CNN models. It can be seen in Figure 36 that the loss value of the GAT and CNN is lower than that of the GCN model, but the stability of the GAT is better. Therefore, the GAT has a certain stability and better accuracy.

According to Table 13, the test set accuracy of the GAT is 100%, which is higher than that of the MLP, Attention, GCN and CNN models.

As can be seen in Table 14, the diagnostic accuracy rate of the GAT for various fault signals can reach 100%, while the diagnosis results of the Attention, MLP, GCN and CNN models are all biased. It can be seen that the diagnostic effect of the graph neural network model is better than that of other models, and the GAT has better stability than the GCN. Therefore, the superiority of the GAT used in this paper is proved via the accuracy and precision rate indexes.

## 5. Conclusions

In this paper, a fault diagnosis model based on the WOA-VMD and GAT was proposed for the identification of fault in rolling bearings with background noise.

The original signal was decomposed using the WOA-VMD, which effectively solved the phenomenon of mode mixing that occurs in traditional modal decomposition. After comparing the noise reduction effects of EMD, EEMD, CEEMD and GA-VMD, the experimental results showed that the root mean square error of the WOA-VMD is 0.213, and the signal-to-noise ratio is 6.912. Thus, the WOA-VMD has the best noise reduction effect.The KNN method was used to construct the graph structure data, and a multi-headed attention mechanism was used to build the GAT rolling bearing fault diagnosis model, which assigned higher weights to the important neighborhoods and improved the sensitivity of the model to graph data containing faults. The diagnostic accuracy of the GAT method was 100%, which was 17.6%, 29.2%, 0.4% and 1.68% higher than that of the MLP, Attention, GCN and CNN models, respectively. This proves that the GAT can achieve more accurate diagnostic decisions for fault data sets.

Although the fault diagnosis algorithm proposed in this study has certain advantages in diagnosis, it is limited to rolling bearings with a constant speed. In the future, the application scope of the research will be extended to rolling bearings with variable speed. In addition, measurements based on physical quantities (as stator current, stray fluxes, thermal images, etc.) will be considered.

## Figures and Tables

**Figure 1 entropy-25-00889-f001:**
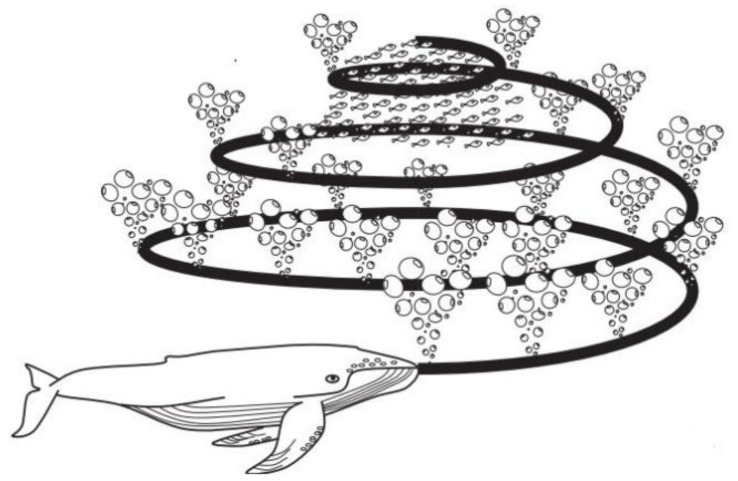
Humpback whales simulate “bubble net” feeding behavior.

**Figure 2 entropy-25-00889-f002:**
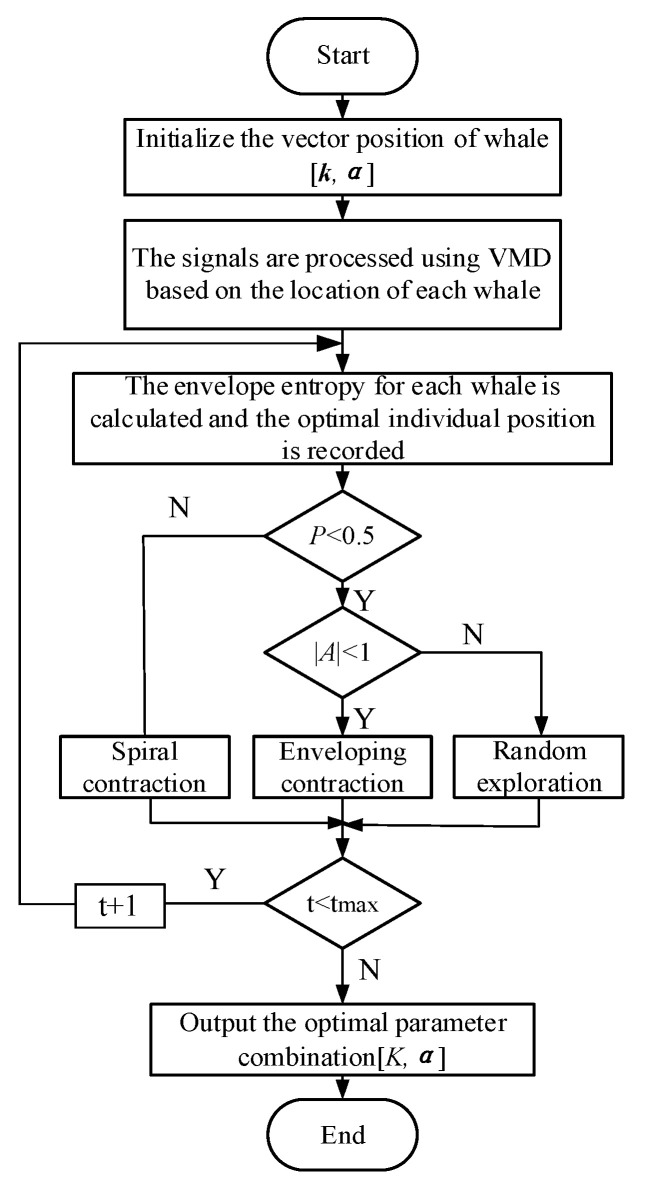
WOA Optimization Process.

**Figure 3 entropy-25-00889-f003:**
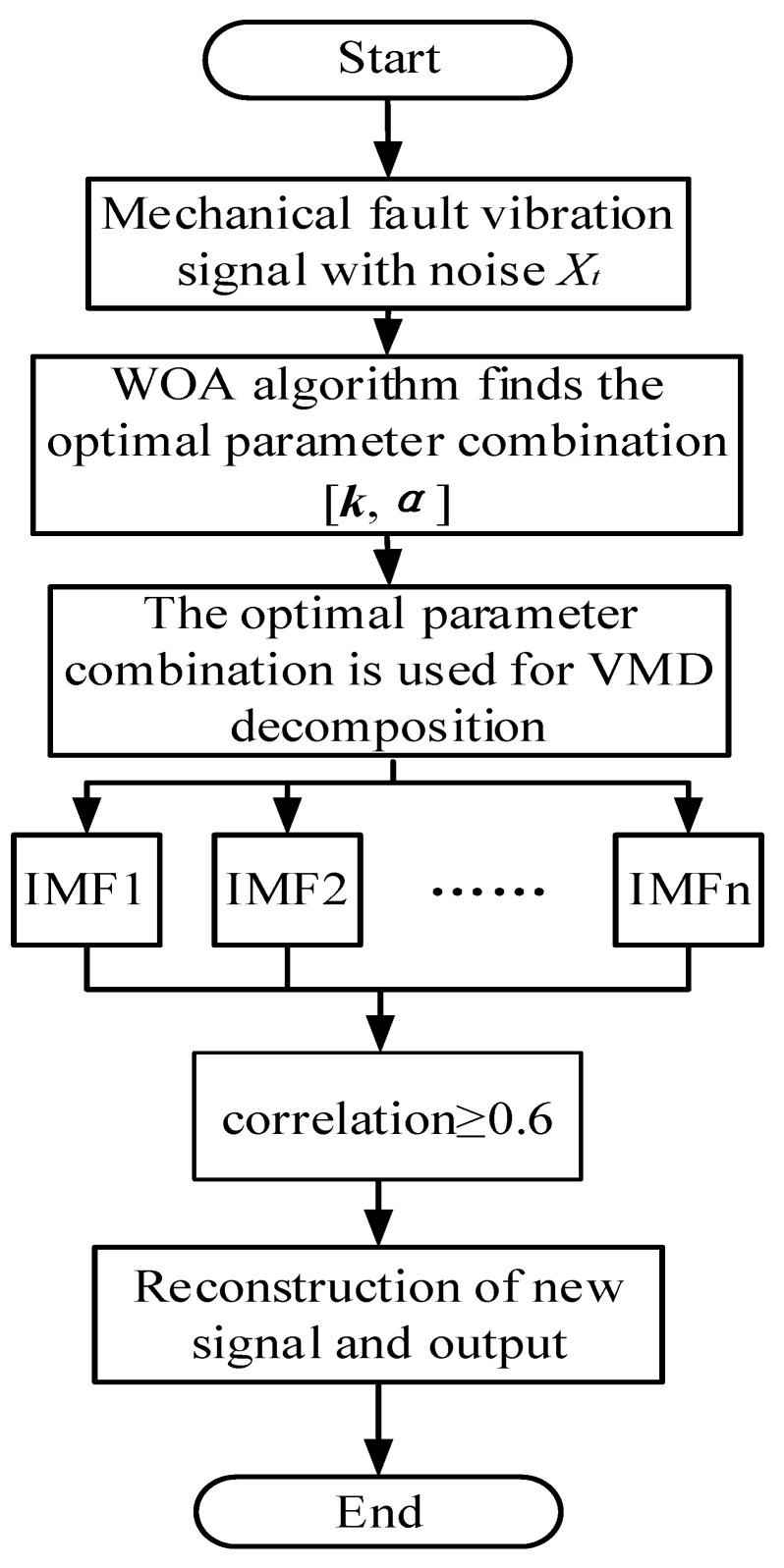
WOA-VMD-based signal noise reduction process.

**Figure 4 entropy-25-00889-f004:**
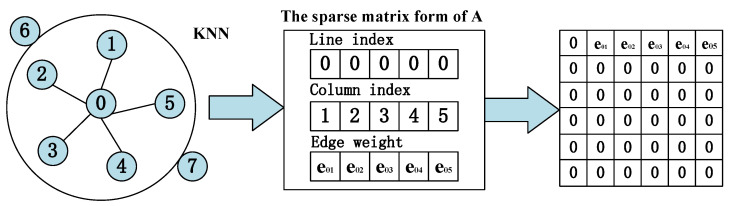
Construct an example of adjacency matrix based on KNN.

**Figure 5 entropy-25-00889-f005:**
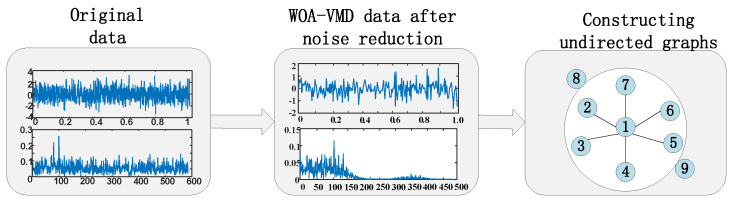
Composition process of the undirected graph of fault data.

**Figure 6 entropy-25-00889-f006:**
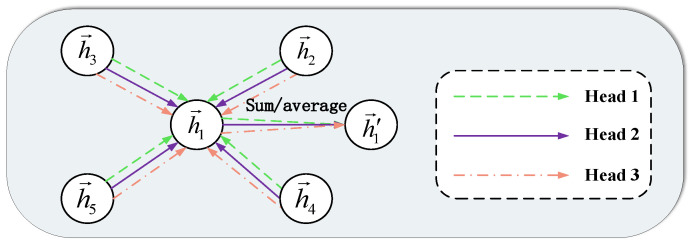
Multi-head attention mechanism schematic diagram.

**Figure 7 entropy-25-00889-f007:**
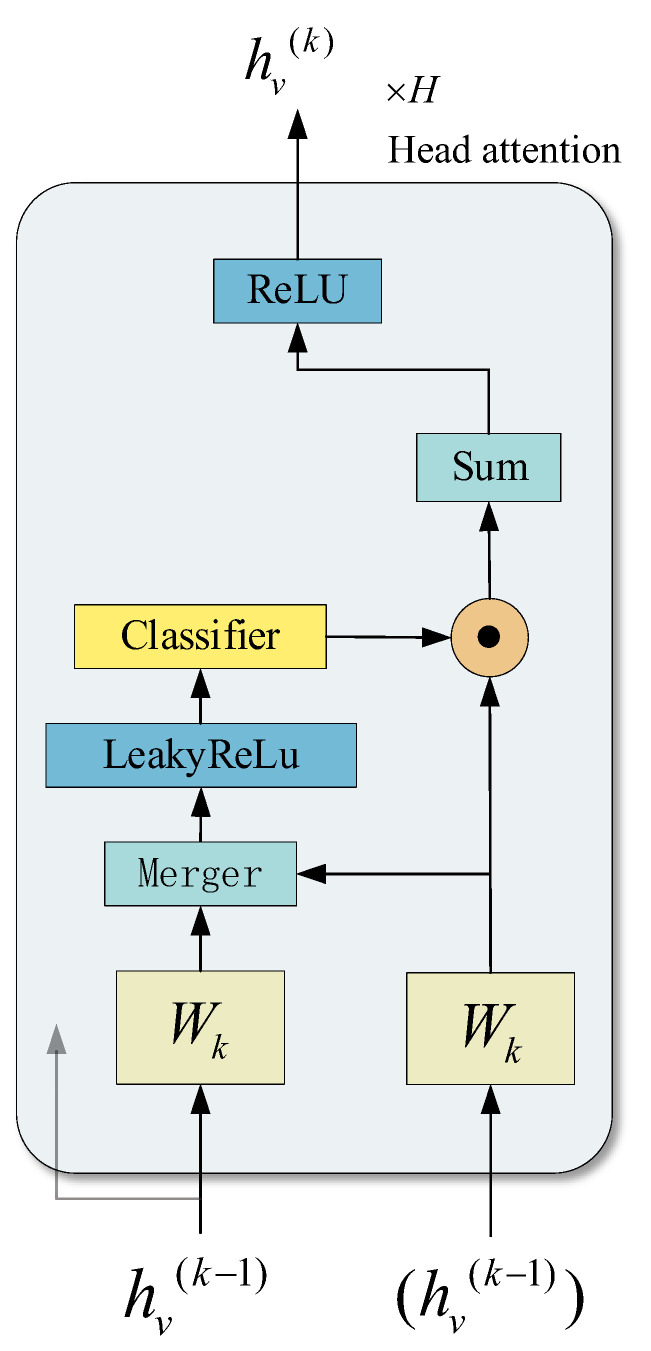
Diagram of GAT model.

**Figure 8 entropy-25-00889-f008:**
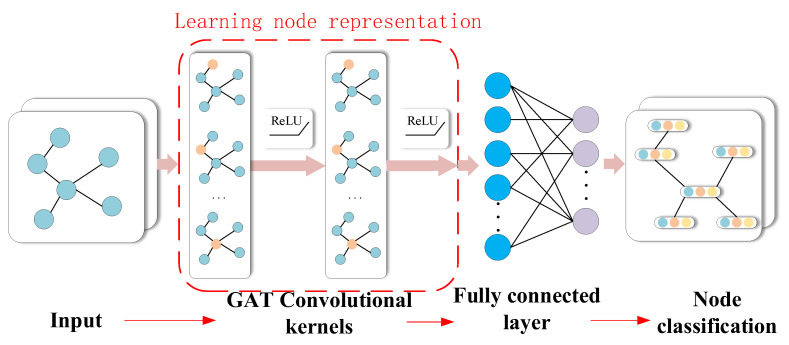
Schematic diagram of GAT layer structure.

**Figure 9 entropy-25-00889-f009:**
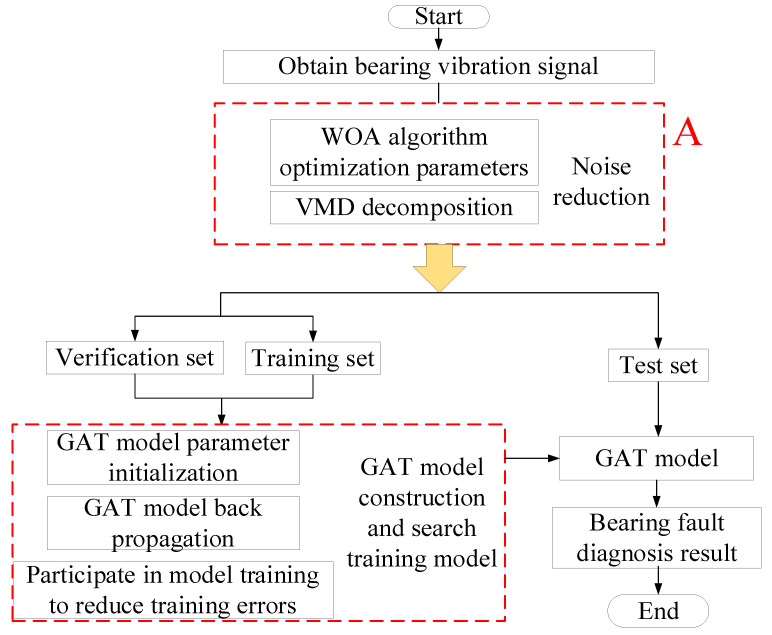
Fault diagnosis flow chart.

**Figure 10 entropy-25-00889-f010:**
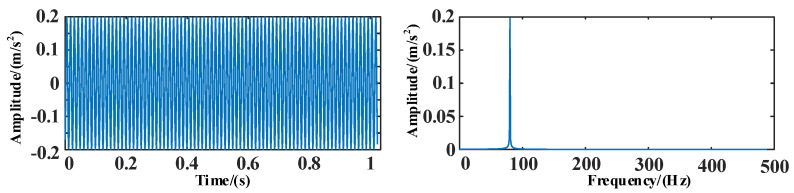
The time and frequency domain diagrams of signal *s*_1_.

**Figure 11 entropy-25-00889-f011:**
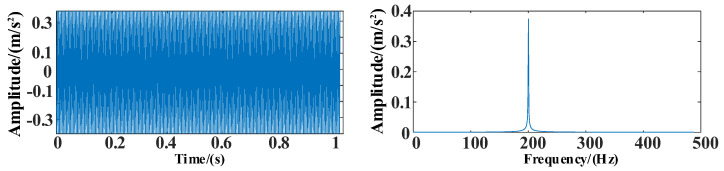
The time and frequency domain diagrams of signal *s*_2_.

**Figure 12 entropy-25-00889-f012:**
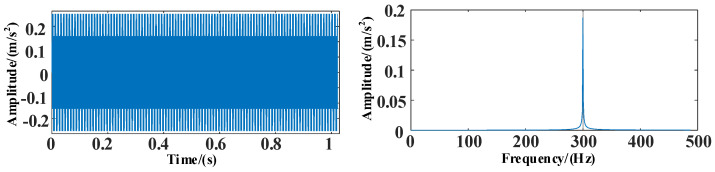
The time and frequency domain diagrams of signal *s*_3_.

**Figure 13 entropy-25-00889-f013:**
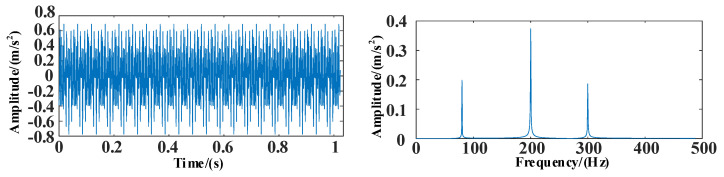
The time and frequency domain diagrams of signal *Z*.

**Figure 14 entropy-25-00889-f014:**
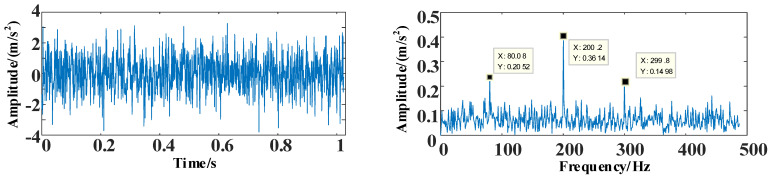
The time and frequency domain diagrams of signal after noise addition.

**Figure 15 entropy-25-00889-f015:**
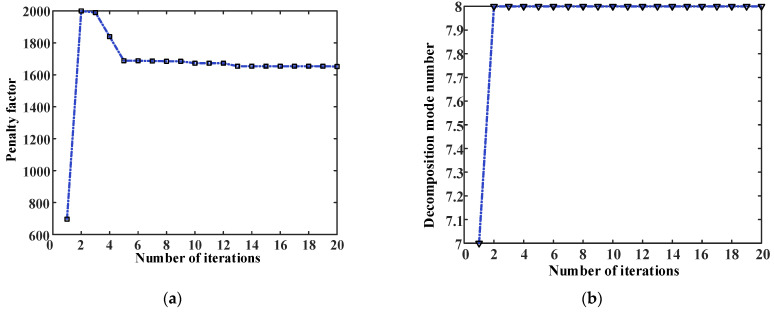
WOA-VMD iteration curve. (**a**) Optimization process curve of penalty factor; (**b**) Optimization process curve of decomposition mode number; (**c**) Envelope entropy iteration curve of WOA.

**Figure 16 entropy-25-00889-f016:**
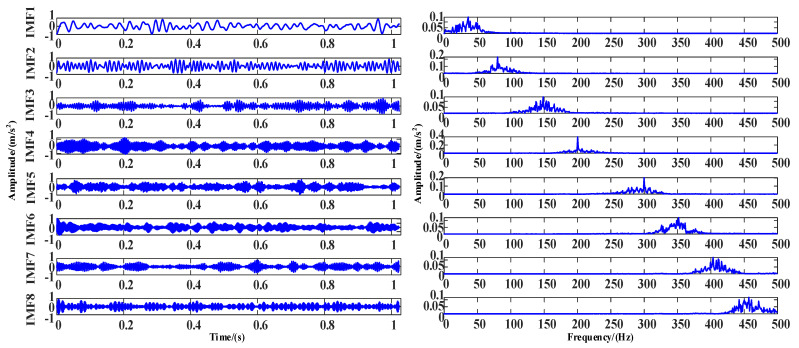
The time and frequency domain diagram of IMF after WOA-VMD decomposition.

**Figure 17 entropy-25-00889-f017:**
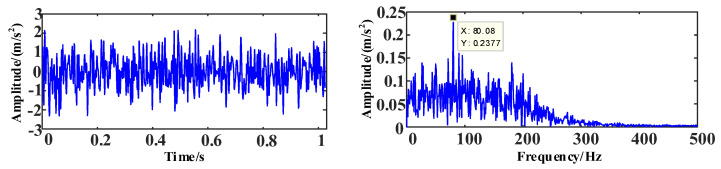
The time and frequency domain diagrams of EMD noise reduction.

**Figure 18 entropy-25-00889-f018:**
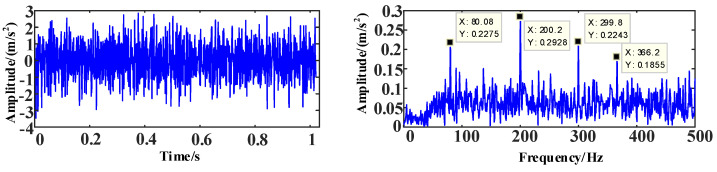
The time and frequency domain diagrams of EEMD noise reduction.

**Figure 19 entropy-25-00889-f019:**
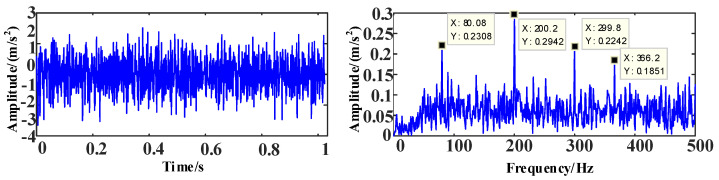
The time and frequency domain diagrams of CEEMD noise reduction.

**Figure 20 entropy-25-00889-f020:**
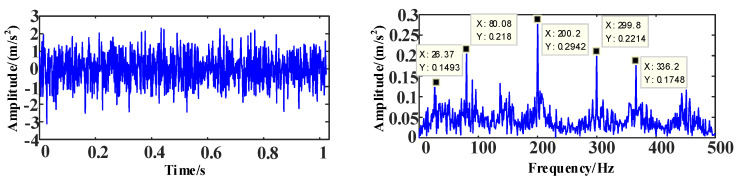
The time and frequency domain diagrams of GA-VMD noise reduction.

**Figure 21 entropy-25-00889-f021:**
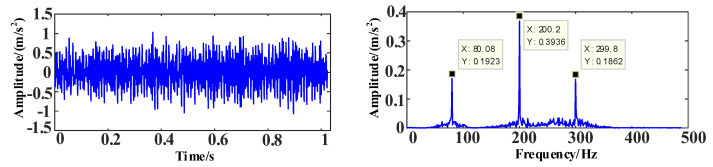
The time and frequency domain diagrams of WOA-VMD noise reduction.

**Figure 22 entropy-25-00889-f022:**
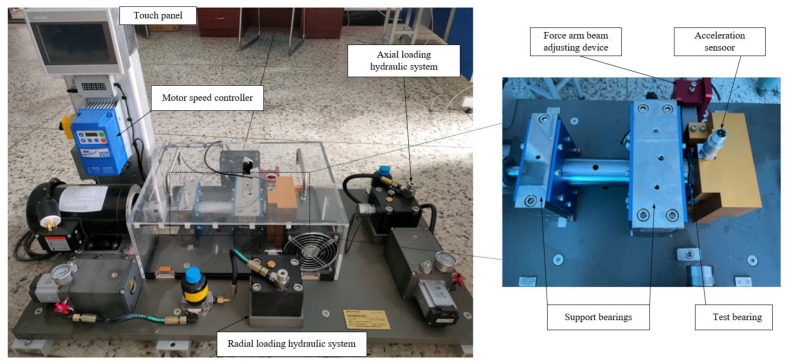
Test bench for rolling bearings.

**Figure 23 entropy-25-00889-f023:**
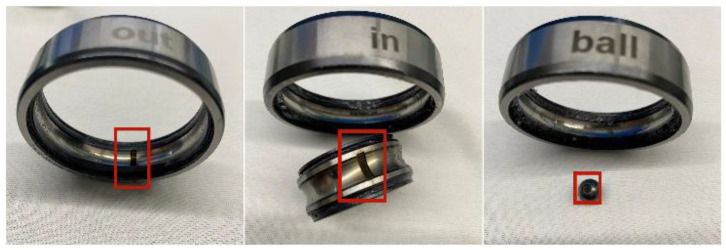
Damage diagrams of rolling bearings.

**Figure 24 entropy-25-00889-f024:**
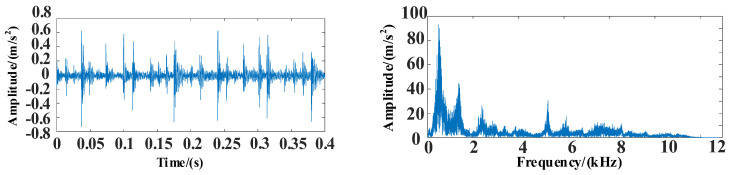
The time and frequency domain diagrams of vibration signals.

**Figure 25 entropy-25-00889-f025:**
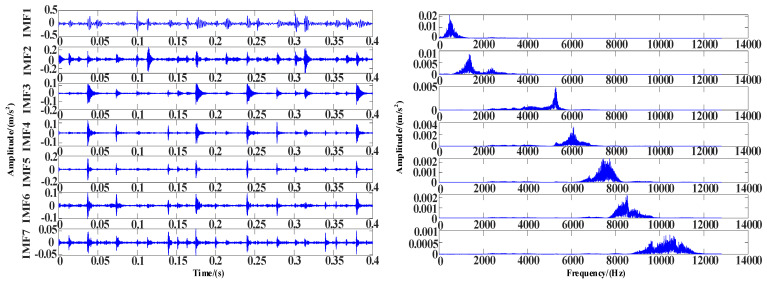
The time and frequency domain diagram of IMF after WOA-VMD noise decomposition.

**Figure 26 entropy-25-00889-f026:**
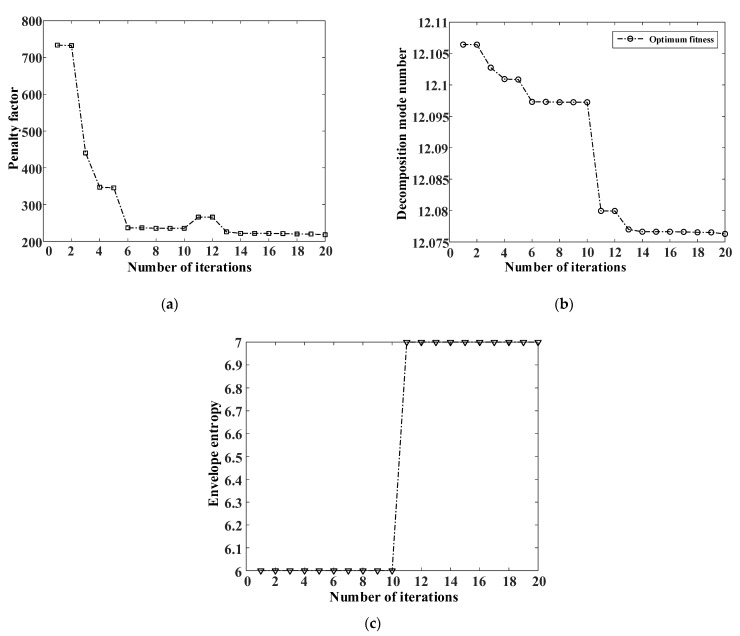
WOA-VMD iteration curve. (**a**) Optimization process curve of penalty factor; (**b**) Optimization process curve of decomposition mode number; (**c**) Envelope entropy iteration curve of WOA.

**Figure 27 entropy-25-00889-f027:**
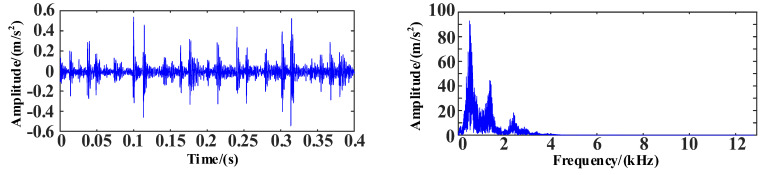
The time and frequency domain diagrams of the signal after WOA-VMD noise reduction.

**Figure 28 entropy-25-00889-f028:**
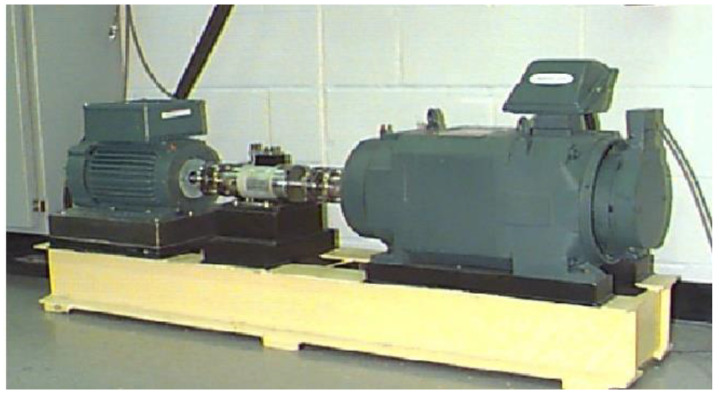
Case Western Reserve University bearing Experimental platform.

**Figure 29 entropy-25-00889-f029:**
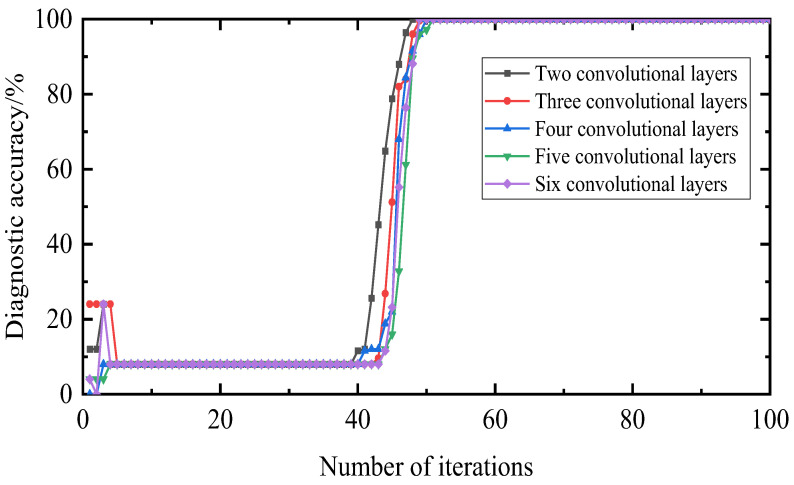
Diagnostic accuracy under different network layers.

**Figure 30 entropy-25-00889-f030:**
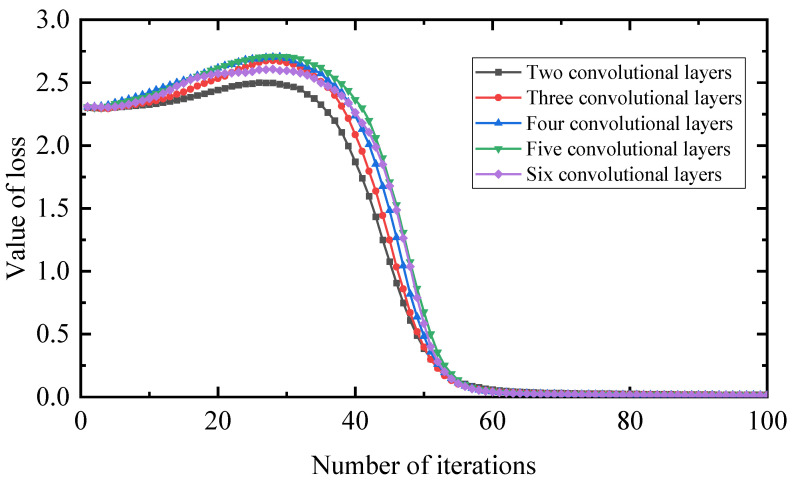
Loss value under different network layers.

**Figure 31 entropy-25-00889-f031:**
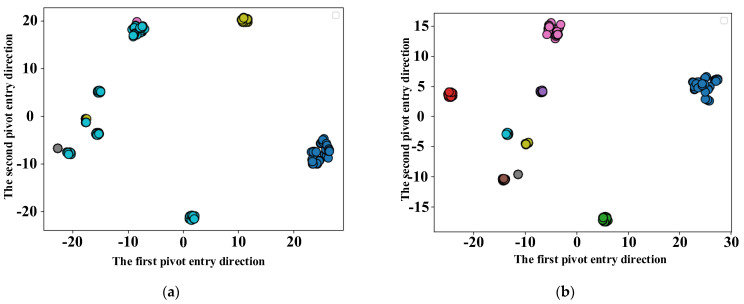
GAT rolling bearing fault diagnosis model visualization. (**a**) Initial data set visualization; (**b**) Visualization of GAT output results.

**Figure 32 entropy-25-00889-f032:**
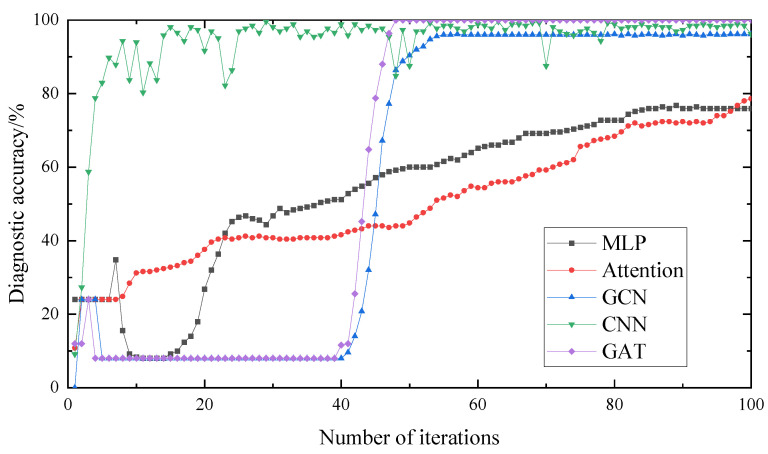
Diagnostic accuracy of different methods.

**Figure 33 entropy-25-00889-f033:**
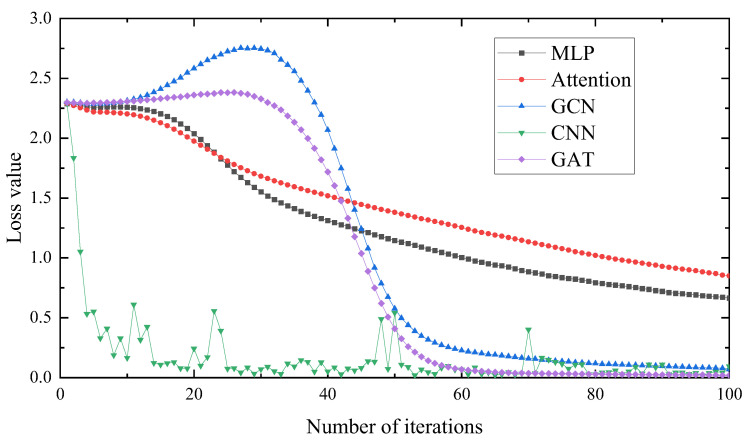
Loss value of different methods.

**Figure 34 entropy-25-00889-f034:**
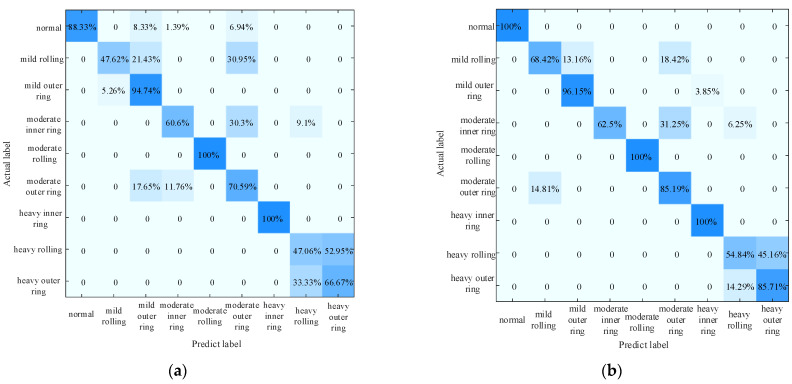
Confusion matrix of different methods. (**a**) Attention; (**b**) MLP; (**c**) GCN; (**d**) GCN; (**e**) GAT.

**Figure 35 entropy-25-00889-f035:**
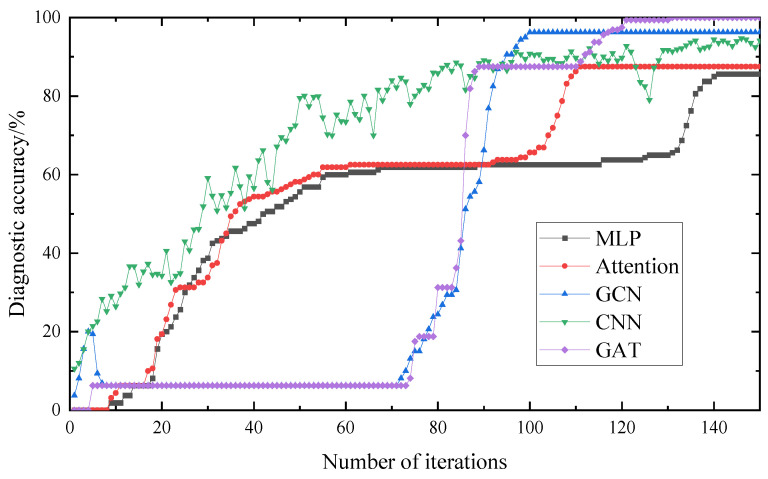
Diagnostic accuracy of different methods.

**Figure 36 entropy-25-00889-f036:**
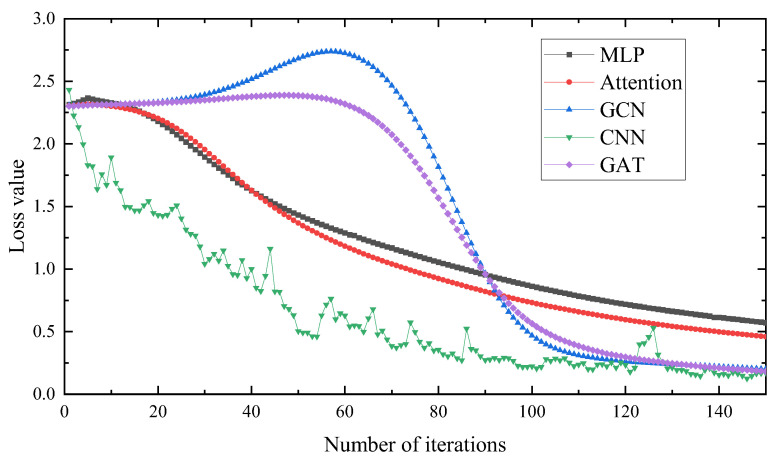
Loss value of different methods.

**Table 1 entropy-25-00889-t001:** Correlation discrimination table.

Interrelation	*r*(*x_t_*,*x_IMF_*) Coefficient Values
Very weakly correlated or uncorrelated	0.0–0.2
weakly correlated	0.2–0.4
Moderate correlation	0.4–0.6
Strongly related	0.6–0.8
Extremely strong correlation	0.8–1.0

**Table 2 entropy-25-00889-t002:** Whale optimization algorithm parameter setting value.

Parameter	Values
Population size	10
Maximum number of iterations	20
Number of variables	2
Range of decomposition layers	[100,3]
Penalty factor range	[2000,7]

**Table 3 entropy-25-00889-t003:** Correlation coefficient value of each IMF.

Mode Component	Correlation Coefficient
IMF1	0.5167
IMF2	0.8005
IMF3	0.4942
IMF4	0.7966
IMF5	0.7854
IMF6	0.6258
IMF7	0.4135
IMF8	0.4638

**Table 4 entropy-25-00889-t004:** Comparison of different noise reduction methods under simulation data.

Noise Reduction Algorithms	Root Mean Square Error	Signal to Noise Ratio
EMD noise reduction	0.738	4.018
EEMD noise reduction	0.780	3.958
CEEMD noise reduction	0.785	3.804
GA-VMD noise reduction	0.702	4.112
WOA-VMD noise reduction	0.213	6.912

**Table 5 entropy-25-00889-t005:** Parameters of the test bench.

Bearing Parameters	Values	Bearing Parameters	Values
Outer ring diameter	51.99 mm	Inner ring diameter	25.40 mm
Weight	0.28 kg	Rolling diameter	7.92 mm
Number of rolling elements	9	Contact angle	0°
Maximum load (static)	7830 N	Maximum load (dynamic)	10,810 kN

**Table 6 entropy-25-00889-t006:** A data set of bearing fault diagnosis experiment.

Radial Loading Force	Fault Location	Data Set	Degree of Damage
0 kg	Inner ring	I_L_0	mild
		I_M_0	moderate
		I_H_0	heavy
	Outer ring	O_L_0	mild
		O_M_0	moderate
		O_H_0	heavy
	Rolling ball	B_L_0	mild
		B_M_0	moderate
		B_H_0	heavy
100 kg	Inner ring	I_L_100	mild
		I_M_100	moderate
		I_H_100	heavy
	Outer ring	O_L_100	mild
		O_M_100	moderate
		O_H_100	heavy
	Rolling ball	B_L_100	mild
		B_M_100	moderate
		B_H_100	heavy
200 kg	Inner ring	I_L_200	mild
		I_M_200	moderate
		I_H_200	heavy
	Outer ring	O_L_200	mild
		O_M_200	moderate
		O_H_200	heavy
	Rolling ball	B_L_200	mild
		B_M_200	moderate
		B_H_200	heavy

**Table 7 entropy-25-00889-t007:** Correlation coefficient value of each IMF.

Mode Component	Correlation Coefficient
IMF1	0.8262
IMF2	0.6075
IMF3	0.2630
IMF4	0.2203
IMF5	0.2069
IMF6	0.1824
IMF7	0.1065

**Table 8 entropy-25-00889-t008:** Experimental parameter settings for different fault states.

Experiment Number	Fault Size	Fault Location	Location of Collection End
No. 1	0	Normal	
No. 2	0.007 inch	Inner ring fault	12 k Driving end
No. 3	0.007 inch	Outer ring fault	12 k Driving end
No. 4	0.007 inch	Rolling fault	12 k Driving end
No. 5	0.014 inch	Inner ring fault	12 k Fan end
No. 6	0.014 inch	Outer ring fault	12 k Fan end
No. 7	0.014 inch	Rolling fault	12 k Fan end
No. 8	0.021 inch	Inner ring fault	48 k Driving end
No. 9	0.021 inch	Outer ring fault	48 k Driving end
No. 10	0.021 inch	Rolling fault	48 k Driving end

**Table 9 entropy-25-00889-t009:** GAT model parameters.

Parameter Name	Value	Parameter Name	Value
Number of the training set sample groups	97	Node deactivation rate	0.2
Number of sample groups of verification set	25	Second fully connected layer	[1024,1024]
Number of test set sample groups	122	Batch normalization	1024
Convolutional kernel of the first layer	[2048,2048]	Loss function	Cross-entropy loss function
Convolutional kernel of the second layer	[2048,2048]	Optimizer	Stochastic gradient descent
Activation function of the first layer	Relu	Training times	100
Activation function of the second layer	Relu	Batch size	64
First fully connected layer	[2048,1024]	Learning rate	0.01

**Table 10 entropy-25-00889-t010:** Diagnostic accuracy of different graph convolution kernel size.

Size	$\Theta \in R^{1024\times 1024}$	$\Theta \in R^{2048\times 2048}$	$\Theta \in R^{4096\times 4096}$
Accuracy	90.40%	96.85%	92.81%
Time/s	42.25	61.58	120.54

**Table 11 entropy-25-00889-t011:** Diagnostic accuracy of test sets of different algorithms.

MLP	Attention	GCN	CNN	GAT
82.40%	70.8%	99.6%	98.32%	100%

**Table 12 entropy-25-00889-t012:** The diagnostic precision of test sets of different algorithms.

	MLP	Attention	GCN	CNN	GAT
normal	100%	83.30%	100%	97.19%	100%
mild rolling	68.42%	47.62%	100%	96.16%	100%
mild outer ring	96.15%	94.74%	100%	100%	100%
moderate inner ring	62.50%	60.60%	100%	95.37%	100%
moderate rolling	100%	100%	100%	88.06%	100%
moderate outer ring	85.19%	70.59%	100%	96.42%	100%
heavy inner ring	100%	100%	100%	97.42%	100%
heavy rolling	54.84%	47.06%	100%	100%	100%
heavy outer ring	85.71%	66.67%	95.24%	98.25%	100%

**Table 13 entropy-25-00889-t013:** Diagnostic accuracy of test sets of different algorithms.

MLP	Attention	GCN	GCN	GAT
85.62%	87.25%	96.25%	94.12%	100%

**Table 14 entropy-25-00889-t014:** Diagnostic accuracy of test sets of different algorithms.

	MLP	Attention	GCN	CNN	GAT
normal	100%	100%	100%	100%	100%
mild rolling	0%	0%	0%	100%	100%
mild outer ring	100%	100%	100%	80%	100%
moderate inner ring	100%	100%	100%	50%	100%
moderate rolling	100%	100%	100%	100%	100%
moderate outer ring	100%	100%	100%	90%	100%
heavy inner ring	100%	100%	100%	100%	100%
heavy rolling	100%	100%	100%	100%	100%
heavy outer ring	0%	0%	100%	100%	100%

## Data Availability

Not applicable.

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
