# Peer review of "A Rolling Bearing Fault Diagnosis Method Based on the WOA-VMD and the GAT"

_entropy, 2023, doi:10.3390/e25060889_

Round 1

Reviewer 1 Report

This paper proposed a rolling bearing fault diagnosis method based on WOA-VMD and the GAT. The performance of the proposed method is validated via experiment data. Some concerns are suggested to be addressed.

1. Eq.(8) is not clear. What is b1, is it different from b? What is arrow[cos(*]? What is 1’ in p<=1’?

2. How are k and a updated in Figure 2?

3. “Mode components with correlation degrees below strong correlation are removed, and then mode components with strong correlation or above are reconstructed to complete noise reduction.” From Table 1, strong correlation and above is achieved when r is larger than 0.6. However, in Figure 3, correlation>=0.3 is selected to reconstruct the signal. This is confused.

4. On Page 6, “Then two key parameters A and B of VMD are determined adaptively using the WOA optimization algorithm to improve the generalization ability of the model.” What are the A and B?

5. The same issue with the above one, “The optimal combination of parameters is output, and the optimal A and B are used to decompose the original signal.” What are the A and B?

The English writing is suggested to be improved.

Author Response

Response to Reviewer 1 Comments

Manuscript ID: entropy-2385974

Title: A Rolling Bearing Fault Diagnosis Method based on the WOA-VMD and the GAT

Dear expert:

We are writing the letter to convey my thanks and my major revisions of your comments. We are honored to be reviewed by your comments. Those comments, which make up for our shortcomings of considering less, are very important for enhancing our paper. All authors have read and approved the manuscript. We have carefully taken the comments into account and responded to each of the points raised by you. Some necessary corrections have been made, and all the altered passages have been highlighted in light yellow. We really hope that our improvements can meet your approval.

Point 1: Eq.(8) is not clear. What is b1, is it different from b? What is arrow[cos(*]? What is 1’ in p<=1’?

Response 1:

Thank expert for pointing out our deficiencies.

The equation in the paper was incorrectly entered and has been changed to the correct formula as shown below

Point 2: How are k and a updated in Figure 2?

Response 2:

Thank expert for pointing out our deficiencies.

The update steps are described below

Step 1: Set the number of whales, the maximum number of iterations and the optimization dimension, and initialize the position information. Set the mode component and penalty factor as  and ;

Step 2: Use the VMD algorithm to decompose the input signal and obtain each IMF function. Calculate the envelope entropy of each IMF according to Equation (11). The envelope entropy was used as a fitness function to find the optimal whale location and retain it;

Step 3: Start the iteration. Generate a random number p in the interval (-1,1). If p < 0.5, it is directly transferred to step 4; otherwise, Equation (8) is used for position update, namely for spiral contraction;

Step 4: Determine the value of |A|. If |A| < 1, update the position type of Equation (5), surrounded by the contraction; otherwise, update the position according to Equation (10), that is, change it to random exploration;

Step 5: Calculate the fitness of each whale and compare it with the previously reserved optimal position. If it is better, replace it with the new optimal solution;

Step 6: Determine whether the iteration is terminated. If , then t=t+1; return to Step 3. Otherwise, the iteration ends and the optimal parameter combination [k, a] is saved.

Point 3: “Mode components with correlation degrees below strong correlation are removed, and then mode components with strong correlation or above are reconstructed to complete noise reduction.” From Table 1, strong correlation and above is achieved when r is larger than 0.6. However, in Figure 3, correlation>=0.3 is selected to reconstruct the signal. This is confused.

Response 3:

Thank expert for pointing out our deficiencies.

An error is indicated in Figure 3, which has been updated as shown below.

Point 4: On Page 6, “Then two key parameters A and B of VMD are determined adaptively using the WOA optimization algorithm to improve the generalization ability of the model.” What are the A and B?

Response 4:

Thank expert for pointing out our deficiencies.

A and B represent the mode component  and the penalty factor , respectively. The original article has been updated to "Then, two key parameters,  and , of the VMD are determined adaptively using the WOA optimization algorithm to improve the generalization ability of the model.".

Point 5: The same issue with the above one, “The optimal combination of parameters is output, and the optimal A and B are used to decompose the original signal.” What are the A and B?

Response 5:

Thank expert for pointing out our deficiencies.

A and B represent the mode component  and the penalty factor , respectively. The original article has been updated to "The optimal combination of parameters is output, and the optimal  and  are used to decompose the original signal.".

This article has undergone English language editing by MDPI.

The above-mentioned major revisions are responses to your comments. Once again, thank you very much for what you have done. Please accept my sincere thanks. Wish you all the best.

Yours truly,

Professor. Wang, Mr. Zhang, Mr. Cao, Dr. Xu, Dr. Fan.

Reviewer 2 Report

Dear Authors,

It was a pleasure to review your paper. Find below some observations regarding it:

1. In the introductory part of your paper, you are considering only the possible fault detection methods based on vibration measurements. There are several other effective methods based on the measurement of other physical quantities (as stator current, stray fluxes, thermal image, etc.), which are not so much sensitive to outer disturbances. See the following papers:

10.1541/ieejjia.7.282

10.1109/ATEE.2013.6563406

10.1109/ASET48392.2020.9118361

You should consider (at least mention) also these possibilities and emphasize the advantages of your detection method also over these.

2. The already included references are relatively recent and their number is suitable. Your text covers the advantages and drawbacks of each considered method. But maybe you should compare them in more depth to clear evidence of the superiority of the method proposed by you.

3. The theoretical description of the proposed method is adequate.

4. The simulation signal used in 4.4.1 is very simple, far from the real (surely noisy) signals that can be measured in the field.

5. As concerning your testbench, a better view of its main part (that under the plexiglass cover) should be mandatorily needed.

6. You obligatory provide a description of the applied measuring equipment, inclusively the precision and type of the used sensors.

7. If you are using the measured results provided by Case Western Reserve University, you should have their permission to use these data and especially to include in your paper the picture of the test bench. You should also provide the reference (website) where the picture of the test bench is taken.

8. When you use non-SI units (like inches) mandatory give also the values in SI units.

9. More details are needed on the way as the detection accuracies were obtained.

10. Correct KHz to kHz

11. To make the conclusion section clearer, you have to include the point-by-point findings of this paper, not only write a simple overview of the paper and make a reiteration of what you had performed and the main results you obtained.

Author Response

Response to Reviewer 2 Comments

Manuscript ID: entropy-2385974

Title: A Rolling Bearing Fault Diagnosis Method based on the WOA-VMD and the GAT

Dear expert:

We are writing the letter to convey my thanks and my major revisions of your comments. We are honored to be reviewed by your comments. Those comments, which make up for our shortcomings of considering less, are very important for enhancing our paper. All authors have read and approved the manuscript. We have carefully taken the comments into account and responded to each of the points raised by you. Some necessary corrections have been made, and all the altered passages have been highlighted in light yellow. We really hope that our improvements can meet your approval.

Point 1: In the introductory part of your paper, you are considering only the possible fault detection methods based on vibration measurements. There are several other effective methods based on the measurement of other physical quantities (as stator current, stray fluxes, thermal image, etc.), which are not so much sensitive to outer disturbances. See the following papers:

10.1541/ieejjia.7.282

10.1109/ATEE.2013.6563406

10.1109/ASET48392.2020.9118361

You should consider (at least mention) also these possibilities and emphasize the advantages of your detection method also over these.

Response 1:

Thank expert for pointing out our deficiencies.

It was added to the introduction "As fault detection is based on physical quantities (such as stator current [4], stray flux [5], thermal image [6], etc.), and because vibration-based measurement methods, lower in cost, easier than direct observation, more sensitive to external interference, and often used in practical engineering, a vibration-based fault detection method is adopted in this paper.". "In addition, measurements based on physical quantities (as stator current, stray fluxes, thermal images, etc.) will be considered." was added to the conclusion.

Kanemaru, M.; Tsukima, M.; Miyauchi, T.; Hayashi, K. Bearing Fault Detection in Induction Machine Based on Stator Current Spectrum Monitoring. IEEJ Journal of Industry Applications. 2018, 7, 282-288.

HarliÅŸca, C.; Szabó, L.; Frosini L.; Albini, A. Diagnosis of rolling bearings faults in electric machines through stray magnetic flux monitoring. In Proceedings of the 2013 8TH International Symposium on Advanced Topics in Electrical Engineering (ATEE), Bucharest, Romania, 23-25 May 2013.

Azeez, A.; Alkhedher M.; Gadala, M. Thermal Imaging Fault Detection for Rolling Element Bearings. In Proceedings of the 2020 Advances in Science and Engineering Technology International Conferences (ASET), Dubai, United Arab Emirates, 04 Feb - 09 Apr 2020.

Point 2: The already included references are relatively recent and their number is suitable. Your text covers the advantages and drawbacks of each considered method. But maybe you should compare them in more depth to clear evidence of the superiority of the method proposed by you.

Response 2:

Thank expert for pointing out our deficiencies.

Thanks to the expert's affirmation, this paper uses WOA to optimize the parameter selection of VMD for noise reduction, and then combines GAT for fault diagnosis. By comparing the noise reduction performance and diagnosis accuracy respectively, the superiority of the proposed method is verified. Further research will be carried out in the future.

Point 3: The theoretical description of the proposed method is adequate.

Response 3:

Thank expert for pointing out our deficiencies.

Thanks for the expert's affirmation.

Point 4: The simulation signal used in 4.4.1 is very simple, far from the real (surely noisy) signals that can be measured in the field.

Response 4:

Thank expert for pointing out our deficiencies.

Although the simulation signal in 4.4.1 is simple, it does not affect the verification of the noise reduction performance and classification effect of the proposed method. At the end of the paper, there are actual noises in the experimental platform, which also verifies the effectiveness of the proposed method.

Point 5: As concerning your testbench, a better view of its main part (that under the plexiglass cover) should be mandatorily needed.

Response 5:

Thank expert for pointing out our deficiencies.

The lab bench picture has been updated and the section under the plexiglass cover has been shown as shown below.

Point 6: You obligatory provide a description of the applied measuring equipment, inclusively the precision and type of the used sensors.

Response 6:

Thank expert for pointing out our deficiencies.

The sensor type is uniaxial acceleration sensor, model ADI150, sensitivity is 100mv/g.

Point 7: If you are using the measured results provided by Case Western Reserve University, you should have their permission to use these data and especially to include in your paper the picture of the test bench. You should also provide the reference (website) where the picture of the test bench is taken.

Response 7:

Thank expert for pointing out our deficiencies.

Permission has been added to the references. The test bench picture is taken from its published paper “Rolling element bearing diagnostics using the Case Western Reserve University data: A benchmark study” which has been cited in the paper.

Case Western Reserve University Bearing Data Center Website. Available online: https://engineering.case.edu/bearingdatacenter/download-data-file (accessed on 22 Apr 2023)

Point 8: When you use non-SI units (like inches) mandatory give also the values in SI units.

Response 8:

Thank expert for pointing out our deficiencies.

In the text "The fault sizes are set to 0.007inch, 0.014inch and 0.021inch. 1 inch=2.54 cm." is added to the value expressed in SI units by non-Si units.

Point 9: More details are needed on the way as the detection accuracies were obtained.

Response 9:

Thank expert for pointing out our deficiencies.

The composition of the experimental platform and details of the experiment have been added " As can be seen in Figure 22, the bearing fault diagnosis test bench consists of a touch panel, motor speed controller, motor, radial loading hydraulic system, ADI150 uniaxial acceleration sensor, axial loading hydraulic system, main shaft, two support 6210 and 18720 bearing, the ER-16K bearing to be measured and a force arm beam adjusting device. The bearing type is ER-16K, and detailed parameters are shown in Table 5. The acceleration sensor was used to obtain the vibration acceleration information of 13 bearing fault states, including 10 single point faults and 3 compound faults (CF). The experimental data were obtained at a sampling frequency of 25.6kHz. A total of 10 groups were collected under each fault state, with each group comprising 32768 sample points." in the paper.

Point 10: Correct KHz to kHz

Response 10:

Thank expert for pointing out our deficiencies.

The x-coordinate units in Figures 24 and 27 have been updated.

Point 11: To make the conclusion section clearer, you have to include the point-by-point findings of this paper, not only write a simple overview of the paper and make a reiteration of what you had performed and the main results you obtained.

Response 11:

Thank expert for pointing out our deficiencies.

The conclusions of this article have been rearranged as follows.

In this paper, a fault diagnosis model based on WOA-VMD and GAT was proposed for the identification of fault in rolling bearings with background noise.

  1. The original signal was decomposed by WOA-VMD, which effectively solved the phenomenon of mode mixing that occurs in traditional modal decomposition. After comparing the noise reduction effects of EMD, EEMD, CEEMD and GA-VMD, the experimental results showed that the root mean square error of WOA-VMD is 0.213, and the signal-to-noise ratio is 6.912. Thus, WOA-VMD has the best noise reduction effect.
  2. The KNN method was used to construct the graph structure data, and a multi-headed attention mechanism was used to build the GAT rolling bearing fault diagnosis model, which as-signed higher weights to the important neighborhoods and improved the sensitivity of the model to graph data containing faults. The diagnostic accuracy of the GAT method was 100%, which was 17.6%, 29.2%, 0.4% and 1.68% higher than that of the MLP, Attention, GCN and CNN models, respectively. This proves that the GAT can achieve more accurate diagnostic decisions for fault data sets.

Although the fault diagnosis algorithm proposed in this study has certain ad-vantages in diagnosis, it is limited to rolling bearings with a constant speed. In the future, the application scope of the research will be extended to rolling bearings with variable speed. In addition, measurements based on physical quantities (as stator current, stray fluxes, thermal images, etc.) will be considered.

The above-mentioned major revisions are responses to your comments. Once again, thank you very much for what you have done. Please accept my sincere thanks. Wish you all the best.

Yours truly,

Professor. Wang, Mr. Zhang, Mr. Cao, Dr. Xu, Dr. Fan.

Round 2

Reviewer 1 Report

The authors have addressed the concerns, and the current form seems good.

Reviewer 2 Report

Thank you for considering my all suggestions.